# Eccentric Muscle Strengthening Using Maximal Contractions Is Deleterious in Knee Osteoarthritis: A Randomized Clinical Trial

**DOI:** 10.3390/jcm13113318

**Published:** 2024-06-04

**Authors:** Emmanuel Coudeyre, Bruno Pereira, Jean-Baptiste Lechauve, Sebastien Girold, Ruddy Richard, Lech Dobija, Charlotte Lanhers

**Affiliations:** 1Service de Médecine Physique et de Réadaptation, CHU Clermont-Ferrand, Université Clermont Auvergne, INRAE, F-63000 Clermont-Ferrand, France; jblechauve@chu-clermontferrand.fr (J.-B.L.); ldobija@chu-clermontferrand.fr (L.D.); clanhers@chu-clermontferrand.fr (C.L.); 2Direction de la Recherche Clinique et de l’Innovation, CHU Clermont-Ferrand, Bâtiment Dunant-3e étage, 58 rue Montalembert, 63003 Clermont-Ferrand, France; bpereira@chu-clermontferrand.fr; 3Institut de Formation en Masso-Kinésithérapie, F-03200 Vichy, France; s.girold@gmail.com; 4Unité de Nutrition Humaine (UNH), CRNH, CHU Clermont-Ferrand, INRAE, F-63000 Clermont-Ferrand, France; rrichard@chu-clermontferrand.fr

**Keywords:** osteoarthritis, knee, eccentric, concentric, rehabilitation

## Abstract

**Objectives:** To show the superiority of eccentric versus concentric strengthening in terms of improving quadriceps strength in knee osteoarthritis (OA), a randomized controlled study was conducted to perform 12 sessions of either eccentric or concentric isokinetic muscle strengthening over 6 weeks. **Methods:** We recruited males and females, aged between 40 and 70 years, with predominantly unilateral femorotibial OA. Exclusion criteria were having a prosthesis, inflammatory arthritis or flare-up of OA, symptomatic patellofemoral OA, cardiovascular or pulmonary disease that could be a contraindication to the study treatment, and any pathology that could cause muscle weakness. The primary endpoint was the between-group difference in change in maximum concentric isokinetic knee extension peak torque (PT) at 60°/s on the OA side at 6 weeks. Secondary endpoints were between-group difference in change in concentric hamstring PT at 60°/s; eccentric quadriceps and hamstring PT at 30°/s; 10 m and 200 m walking speeds; pain and functional status (WOMAC score) at 6 weeks and 6 months. **Results:** The sample consisted of 11 females and 27 males, with a mean age of 57.7 ± 7.52 years and a body mass index (BMI) of 25.95 ± 3.93 kg/m^2^. Quadriceps strength increased more at 6 weeks in the concentric than the eccentric group with no statistical difference. There was a rate of 25% major adverse events in the eccentric group. **Conclusions:** Eccentric training resulted in a smaller improvement in quadriceps strength than concentric training and was associated with a high risk of muscle injury, particularly to the hamstring muscles.

## 1. Introduction

Osteoarthritis (OA) affects an estimated 10 million people, 4.6 million of whom have symptomatic OA [1]. This figure is expected to increase with the aging of the population [2]. Osteoarthritis has an undeniable impact on functional capacity, which places it among the leading causes of disability in Western societies [3]. The knee is the second most frequent symptomatic OA site, with the medial femorotibial compartment affected in more than 2/3 of cases [4,5]. This is associated with a loss of lower limb muscle strength, particularly of the quadriceps muscle. This muscle weakness is not only a cause of pain and instability, but also contributes to the disease progression [6]. The degree of quadriceps muscle strength loss also appears to correlate with the intensity of pain and the extent of the functional disability [7,8,9]. Conversely, people with higher levels of strength have less pain and less functional disability [10]. The value of physical exercise and muscle strengthening in OA has been demonstrated [3,11]. 

Isokinetic devices provide reliable, reproducible, and validated measurements of force in a dynamic configuration, which is more functional than measurements obtained during isometric contractions. They can also be used to provide precise training programs. Isokinetic training produces contractions at a constant speed, using a dynamometer whose resistance is self-adapted to the force developed by the individual [9]. Several studies have demonstrated the benefits of isokinetic strengthening in the management of knee OA [10,11,12,13,14]. In addition to its effectiveness in improving muscle strength and function, eccentric isokinetic training has advantages over concentric training in terms of cardiovascular tolerance [15]. However, studies of good methodological quality comparing isokinetic rehabilitation to other validated rehabilitation techniques are relatively rare, making it impossible to draw definitive conclusions regarding the most effective methods [16]. Eccentric (ECC) muscle performance, particularly of the quadriceps, plays an essential role in activities of daily living such as walking and descending stairs. The gain in muscular strength after ECC training is reported to be greater than after concentric (CONC) training [17], with a lower energy cost [18,19,20]. In addition, isokinetic dynamometers can be adapted to the motor performance and tolerance of the person, thus limiting the occurrence of injury in people with impaired physical capacities [21].

The main aim of this study was to show the superiority of 12 sessions of ECC exercise over 12 sessions of CONC exercise in improving quadriceps muscle strength in people with femorotibial OA. The secondary objectives were to evaluate isokinetic muscle strengthening in ECC mode, pain reduction and functional improvement in the parameters of walking (maximum speed over 10 and 200 m), functional status of the patient using the Western Ontario and MacMaster Universities osteoarthritis index [22] and the adherence.

## 2. Materials and Methods

### 2.1. Design

We conducted a randomized controlled trial. Participants were randomly allocated to 1 of 2 groups (ECC or CONC training) in a 1:1 ratio. The randomization list was generated by block randomization and performed automatically using Stata software (version 15, StataCorp, College Station, TX, USA) by an independent statistician after verification of eligibility and signing of informed consent. This study was approved by the CPP SUD EST VI (No 2011-A00894-37) ethical committee, and all participants provided written consent for participation. This study was registered on clinical trials CT NCT01586130. The results are reported in accordance with the CONSORT guidelines for non-pharmacological trials (Table A1).

### 2.2. Participants

Participants were recruited from Clermont-Ferrand University Hospital Rehabilitation Medicine department. Outpatients followed in consultation for knee OA were included. The inclusion criteria were males and females aged between 40 and 75 years with unilateral, medial, femorotibial OA of moderate radiological stage (graded 1–3 according to the Kellgren and Lawrence classification), with no contraindications to the training protocols evaluated. Exclusion criteria were having a prosthetic knee, inflammatory arthritis, or flare-up of OA, symptomatic patellofemoral OA, cardiovascular or pulmonary disease that could contraindicate performance of the training protocols, and any pathology that could cause muscle weakness.

During the inclusion visit, participants underwent a general clinical examination, an electrocardiogram, and additional examinations in case of cardiac abnormality (stress test and cardiological assessment). Baseline outcomes were also measured (see Outcomes section below). The initial maximal moment of force was measured using the Cybex HumacNorm^®^ isokinetic dynamometer. Patients familiarized themselves with testing procedures by performing three consecutive warmup trials for each muscle group and speed. During tests, the subjects performed five maximal continuous flexion–extension repetitions of both legs at each angular velocity: A 1 min rest was allowed between each contraction speed. A 5 min rest was allowed between legs. The best repetition among 5 was kept.

This measurement was used to determine the initial training intensity.

### 2.3. Interventions

The interventions were performed by a team of physiotherapists from the Clermont-Ferrand University hospital who were not blinded to group allocation.

The interventions consisted of 12 sessions of quadriceps and hamstring muscle strengthening of the OA knee using the Cybex HumacNorm^®^ isokinetic dynamometer, which means that the patients were seated with 110° hip flexion and 90° knee flexion. It is secured to the back of the chair by a lap and shoulder belt. The thigh homolateral to the tested knee is secured to the seat by a strap. The dynamometer’s center of rotation is positioned opposite the lateral condyle. The device for attaching the leg to the dynamometer, including an anti-spin module, is positioned just below the tibial tuberosity. It allows a dynamic voluntary muscular contraction to take place at a constant angular speed thanks to a self-adapting resistance. It allows a dynamic voluntary muscular contraction to take place at a constant angular speed thanks to a self-adapting resistance.

The exercises were either exclusively eccentric (intervention group) or exclusively concentric (control group). Sessions were performed twice per week over 6 weeks and had to be at least 48 h apart.

Each session began with a warm-up on a static bike for 5 min at low rate and low load. The concentric group performed 3 sets of 10 repetitions of extension/flexion at adapted effort at a speed of 60°/s, for a total of 30 repetitions. The eccentric group performed 3 sets of 10 repetitions of extension/flexion at a speed of 30°/s for a total of 30 repetitions. Each repetition was followed by a 20 s rest and 1 mn between sets.

The intensity was increased progressively over the sessions. The 1st session was performed at 60% of the initial maximal moment of force, and the intensity was increased by 10% at each session, up to the 6th session if tolerated. In the event of fatigue or muscle pain during a session, the intensity was set at the level of the previous session. We therefore chose to quantitatively progress the exercises by asking participants to develop maximal torque with no pain at each repetition. 

Participants were instructed to develop a maximum force torque without pain on each repetition, and visual on the PC screen and standardized verbal feedback were provided to increase motivation (Table A2).

### 2.4. Outcomes

Outcomes were measured at baseline, 6 weeks (short-term assessment, directly at the end of the training sessions) and at 6 months (medium-term assessment). The same physician who was blinded to group allocation performed all the assessments.

The primary endpoint was the change in concentric isokinetic knee extension peak torque (PT) at 60°/s on the OA side from baseline to 6 weeks.

Secondary endpoints were changed in the following: concentric hamstring PT at 60°/s; eccentric quadriceps and hamstring PT at 30°/s; 10 m and 200 m walking speeds; pain (VAS rating); and WOMAC score from baseline to 6 weeks and baseline to 6 months.

Pain was assessed using a visual analogue scale (VAS) from 0 to 100 before and after each session.

Adherence was measured by the number of sessions performed (maximum 12).

The WOMAC osteoarthritis index, a self-rated questionnaire validated for lower limb OA, was used to grade the impact of the OA on the person’s health status. The index is composed of 24 items grouped into 3 dimensions: pain (A), stiffness (B), and functional capacity (C). 

### 2.5. Statistics

The sample size estimation was estimated for a 15% greater increase in the muscle strength of the quadriceps of the OA knee in the ECC group than the CONC group, with an SD of 20%, according to the results of Huang et al. [10] and Tuzun et al. [11] for concentric training. Therefore, for a two-sided type I error of 5% and a statistical power at 90%, 40 participants were required per group. The rate of losses to follow-up for the primary endpoint after treatment was considered negligible [11]. The high number of major adverse events motivated the promotor of study to carry out an interim analysis to decide if this study should be continued.

The continuous variables are expressed by mean and SD or median and interquartile range, according to the normality of their distribution (verified using the Shapiro–Wilk test). The primary analysis was conducted by intention-to-treat with the last observation carried forward imputation approach for missing data (because the primary outcome was not measured or because the person stopped this study due to an adverse event). A per-protocol analysis was also carried out for participants with no missing data for the primary endpoint at 6 weeks. Between-group comparisons of change in the quadriceps and hamstring PT at 6 weeks and 6 months and disability at 6 weeks and 6 months were performed with the Student *t*-test or the Mann–Whitney test if the assumptions for the *t*-test were not met. The homoscedasticity was analyzed using the Fisher–Snedecor test. The results were expressed using effect-sizes and 95% CI. Between-group comparisons of categorical variables were performed with the Chi-squared or Fisher’s exact test. Multivariate analysis was conducted to adjust the post-intervention results on the WOMAC score and the initial maximal moment of force assessed at inclusion. All statistical analyses were performed using Stata software (version 15, StataCorp, College Station, TX, USA) for a two-sided type I error at 5%. Because of the potential for type 1 error due to multiple comparisons, findings from the secondary endpoint analyses were considered as exploratory.

### 2.6. Ethics

Participant consent was obtained in writing and this study was approved by the CPP SUD EST VI (No 2011-A00894-37). This study has been registered on clinical trials CT NCT01586130.

## 3. Results

### 3.1. Sample

Forty people were included between February 2012 and August 2014. Two withdrew consent before randomization; therefore, 38 were randomly allocated to the ECC or CONC groups (Figure 1). Demographics and baseline data did not differ between groups (Table 1). The sample consisted of 11 females and 27 males, with a mean age of 57.7 (7.52) years and a body mass index (BMI) of 25.95 (3.93) kg/m^2^. 

### 3.2. Adverse Events and Adherence

Four major adverse events that justified withdrawal of the participant from the training program occurred; all five were in the ECC group. Four occurred during the muscle strengthening sessions and one at the 6-week assessment. We did not perform any additional exam to the five patients as they were not able to continue the training due to muscle lesion, which was a clinical confirmation without MRI or plain films.

The rate of adverse events in the ECC group was thus 26.3%. In contrast, no adverse events occurred in the CONC group (*p* = 0.046). In total, 13/15 (86%) of participants in the CONC group and 13/17 (76.5%) in the ECC group (*p* = 0.66) completed all sessions (Table 2).

### 3.3. Primary Outcome

At 6 weeks, the change in concentric knee extension PT was greater in the CONC than the ECC group, although the difference was not significant (CONC: +28 Nm/kg; 95% CI, −17 to 55; ECC 10 Nm/kg; −100 to 52) (*p* = 0.34, ES = 0.32; −0.32 to 0.96) (Table 3 and Figure 2). The per-protocol analysis confirmed these findings with *p* = 0.97 (ES = 0.14; −0.58 to 0.85) (CONC 37 Nm/kg; 95% CI −3 to 66; ECC 20 Nm/kg; 10 to 61). Concentric quadriceps PT was significantly higher in the CONC group (*p* = 0.04, ES = 0.67; 95% CI 0.01 to 1.32). Multivariate analysis confirmed the findings (*p* = 0.04).

### 3.4. Secondary Outcomes

None of the secondary outcomes differed between groups at 6 weeks or 6 months (Table 4). We found no significant difference between the two groups for pain and antalgics consumption at baseline: 8/19 (42.1%) for the concentric group vs. 11/19 (57.9%) for the eccentric group.

## 4. Discussion

This randomized controlled trial did not find a short-term (6 weeks) superiority of ECC muscle training over CONC training on concentric quadriceps muscle strength in people with knee OA, in contrast with our hypothesis. Furthermore, tolerance of the ECC contraction mode was very poor. A high proportion (12.5%) of major adverse events occurred, which justified the withdrawal of the participants concerned and resulted in premature cessation of this study at the half of the inclusions.

Unmasking revealed that these events occurred exclusively in the ECC group. Four events occurred during the muscle strengthening sessions and the fifth occurred during the 6-week evaluation. Four hamstring muscle lesions occurred: a stage 3 myoaponeurotic lesion of the biceps femoris according to the classification of Rodineau and Durey [23] and three elongation-type lesions. The fifth adverse event was an intolerance of eccentric work which occurred progressively across the sessions and prevented continuation. In contrast, no such major adverse events occurred in the CONC group. In addition, hamstring strength reduced by around 20% at 12 weeks and 6 months following the ECC training. Therefore, the main result of this study is that maximum eccentric hamstring contraction strengthening training comports a risk of injury in people with knee OA. 

Preventive measures had been put in place to limit muscle injuries. Exercises, particularly eccentric, are known to cause delayed onset muscle soreness (DOMS) [24]. DOMS generally occurs 8 h after the end of the exercise and includes a combination of pain, stiffness, loss of active mobility, and reduction in strength. The peak of the symptomatology is normally reached at 48 h and then gradually fades over a week. Therefore, if DOMS occurred during the first training sessions, especially in this sample of non-athletic individuals, it was expected to disappear as the sessions progressed [25]. In addition, a minimum delay of 48 h between training sessions and 1 week between the last training session and the second evaluation was observed to minimize the impact of DOMS on performance. Furthermore, the exercises were progressed over the sessions by increasing the number of repetitions, as is more often the case in recent studies on the subject [10]. The Cybex Humac Norm^®^ dynamometer did not allow an exercise intensity to be set relative to the moment of maximum force. 

The short- and medium-term analyses showed that both CONC and ECC contraction modes improved quadriceps strength, in line with the data in the literature [26]. The beneficial effect of concentric strengthening on quadriceps muscle strength had already been shown [26]. Strength also increased on the non-trained side. This “cross effect” phenomenon of muscle strengthening is well known and is in line with the data in the literature [27,28,29].

Regarding the self-evaluation of function using the WOMAC questionnaire, there was no statistically significant difference between the two groups. The participants included in this study had early-stage OA; therefore, the impact and change in functional level was not very marked. With regards to pain intensity at inclusion, there was no between group difference. VAS pain ratings are relatively reliable and reproducible; it is therefore unlikely that the initial maximal strength evaluation was affected by pain.

### Limitations

With regards to the primary outcome, we chose to evaluate concentric, rather than eccentric, strength before and after the strengthening protocol because of its preferential and validated use as a measure of muscle strength and good reproducibility of results at a slow speed of 60°/s. Although muscles more often work eccentrically during activities of daily living, such as walking, which is particularly impaired in OA, we did not evaluate eccentric strength because it is less reproducible and there are fewer data in the literature on this subject [10,11].

To avoid an evaluation bias related to this choice of primary endpoint, a statistical cross-tabulation analysis was performed. This analysis measured change in concentric strength if the participant was randomized to the CONC group and in eccentric strength if the participant was randomized to the ECC group. These results also supported the superiority of concentric training at both the short- and medium-term assessments.

As this study was based on an intervention using a medical device, double blinding was not possible: the participant and physiotherapist had to be aware of the type of strengthening performed. However, the evaluator was blinded to group allocation. Participants in both groups were instructed not to discuss their rehabilitation sessions during the evaluations. 

The interest of dynamic work in ECC mode in this population should not be excluded in this pathology. Indeed, the occurrence of these injuries is most probably related to the maximal character of the muscle contraction. In the future, it would be interesting to analyze the effectiveness of ECC strengthening protocols that are qualitatively adjustable, and thus avoid this type of maximal contraction.

We did not practice other biologic exams like CRP to see the impact of exercise, as increased muscle strength was the primary endpoint.

Patients were free to take paracetamol or non-steroidal anti-inflammatory drugs on request, but this was not monitored during this study.

## 5. Conclusions

Eccentric muscle strengthening using maximal isokinetic contractions is deleterious in people with knee OA. Muscle strength increased less with eccentric than concentric training and eccentric training was associated with a high risk of muscle injury in this sample, particularly for the hamstring muscles. The occurrence of adverse events in the eccentric training group led to the premature termination of this study. However, dynamic, concentric muscle strengthening is of value in people with knee OA, since it allows significant improvements in strength and function in both the short and medium term.

## Figures and Tables

**Figure 1 jcm-13-03318-f001:**
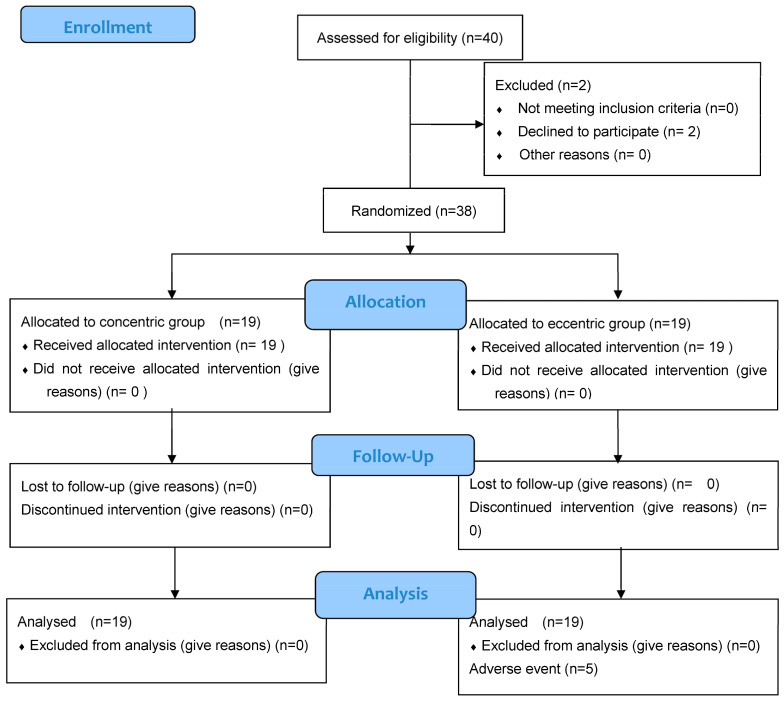
Flow Chart.

**Figure 2 jcm-13-03318-f002:**
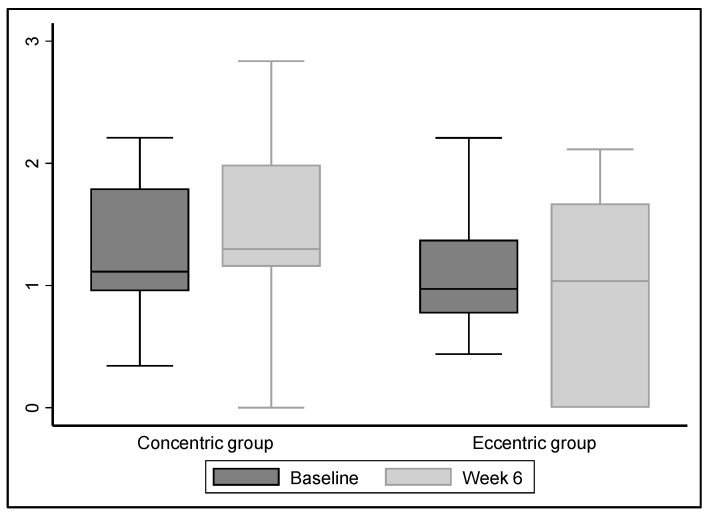
Quadriceps peak torque at 6 weeks.

**Table 1 jcm-13-03318-t001:** Baseline sample characteristics.

	Concentric Group (n = 19)	Eccentric Group (n = 19)	Total (N = 38)
Sex (male)	13 (68.4)	14 (73.7)	27 (71.1)
Age (years)	55.9 ± 7.3	59.7 ± 7.6	57.8 ± 7.6
BMI (kg/m^2^)	25.2 ± 3.9	26.3 ± 3.9	25.8 ± 3.9
Kellgren–Lauwrence score = 1	6 (31.6)	5 (26.3)	11 (29.0)
Kellgren–Lauwrence score = 2 + 3	11 (57.9)	13 (68.4)	24 (63.2)
Knee flexion (injured) Knee extension (injured)	136.6 ± 6.5 −2.5 ± 3.5	132.6 ± 7.4 −3.8 ± 4.4	134.6 ± 7.1 −3.2 ± 4.0
Knee flexion (healthy) Knee extension (healthy)	140.8 ± 7.2 −0.5 ± 1.4	139.2 ± 5.2 −0.6 ± 2.3	140.0 ± 6.2 −0.6 ± 1.9
VAS pain (OA) (/100)	25.3 ± 17.1	35.5 ± 26.6	30.5 ± 22.8
VAS pain (non-OA) (/100)	12.9 ± 9.6	11.7 ± 19.2	12.3 ± 15.1
PT OA knee (Nm/kg):			
Quadriceps concentric	1.29 ± 0.47	1.07 ± 0.44	1.18 ± 0.46
Hamstring concentric	0.84 ± 0.32	0.76 ± 0.29	0.80 ± 0.31
Quadriceps excentric	2.04 ± 0.66	1.81 ± 0.70	1.92 ± 0.69
Hamstring excentric	1.38 ± 0.49	1.29 ± 0.61	1.33 ± 0.54
PT non-OA knee (Nm/kg):			
Quadriceps concentric	1.37± 0.44	1.22 ± 0.43	1.29 ± 0.44
Hamstring concentric	0.89 ± 0.25	0.82 ± 0.28	0.85 ± 0.26
Quadriceps excentric	2.22 ± 0.76	2.03 ± 0.77	2.12 ± 0.76
Hamstring excentric	1.29 ± 0.35	1.25 ± 0.50	1.27 ± 0.42
WOMAC A pain (/20)	5.2 ± 2.6	6.8 ± 3.6	6.0 ± 3.2
WOMAC B disability (/68)	13.7 ± 8.7	22.7 ± 13.1	18.2 ± 11.9
WOMAC C stiffness (/8)	2.7 ± 1.5	3.8 ± 1.9	3.3 ± 1.8
10 m walk speed (m/s)	1.97 ± 0.19	1.90 ± 0.28	1.93 ± 0.24
200 m walk speed (s)	112.9 ± 17.3	119.7 ± 15.8	116.3 ± 16.7

BMI: Body Mass Index; VAS: visual analog scale; PT: peak torque (Nm/Kg); WOMA data are mean ± SD.

**Table 2 jcm-13-03318-t002:** All adverse events.

Patient	Randomization	Events
#5	concentric group	PREMATE STOP
#25	concentric group	PREMATE STOP
#36	concentric group	INJURY M6
#8	eccentric group	PREMATE STOP
#10	eccentric group	INJURY W6
#23	eccentric group	INJURY W6
#26	eccentric group	INJURY W6
#31	eccentric group	INJURY W6
#35	eccentric group	INJURY W6

**Table 3 jcm-13-03318-t003:** Change (%) in quadriceps and hamstring peak torque at 6 weeks and 6 months.

	Concentric Group	Eccentric Group	*p*-Value
**Concentric OA knee**			
**Quadriceps**			
Baseline	1.29 ± 0.47	1.07 ± 0.44	
At W6	1.41 ± 0.77	0.94 ± 0.70	
**Change at W6 (primary endpoint)**	**28 [−17 to 55]**	**10 [−100 to 52]**	**0.34**
At M6	1.26 ± 0.99	1.03 ± 0.83	
Change at M6	23 [−99 to 74]	16 [−100 to 74]	0.74
Hamstring			
Baseline	0.84 ± 0.32	0.76 ± 0.29	
At W6	0.83 ± 0.42	0.59 ± 0.39	
Change at W6	14 [−12 to 44]	−1 [−100 to 24]	0.21
At M6	0.62 ± 0.48	0.59 ± 0.48	
Change at M6	−13 [−99 to 36]	−3 [−100 to 29]	0.96
Ratio Hamstring/Quadriceps			
Change at W6	−10 [−26 to 3]	−13 [−100 to 0]	0.42
Change at M6	−39 [−99 to −16] *	−47 [−100 to −11] *	0.96
**Eccentric OA knee**			
Quadriceps			
Baseline	2.04 ± 0.66	1.81 ± 0.70	
At W6	1.91 ± 1.04	1.41 ± 1.02	
Change at W6	0 [−24 to 30]	0 [−100 to 22]	0.65
At M6	1.44 ± 1.11	1.43 ± 1.14	
Change at M6	−12 [−99 to 6] *	−1 [−100 to 38]	0.59
Hamstring			
Baseline	1.38 ± 0.49	1.29 ± 0.61	
At W6	1.12 ± 0.66	0.86 ± 0.68	
Change at W6	−17 [−45 to 0]	−13 [−100 to 13] *	0.99
At M6	0.74 ± 0.64	0.78 ± 0.68	
Change at M6	−40 [−99 to 16] *	−40 [−100 to 20] *	0.85
Ratio Hamstring/Quadriceps			
Change at W6	−8 [−39 to 2]	−20 [−100 to 0]	0.39
Change at M6	−35 [−99 to 12] *	−32 [−100 to 0] *	0.83
**Concentric–eccentric OA knee**			
Quadriceps			
Change at W6	28 [−17 to 55]	0 [−100 to 22]	0.04
Change at M6	23 [−99 to 74]	−1 [−100 to 38]	0.22
Hamstring			
Change at W6	14 [−12 to 44]	−13 [−100 to 13] *	0.04
Change at M6	−13 [−99 to 36]	−40 [−100 to 20] *	0.42
Ratio Hamstring/Quadriceps			
Change at W6	−10 [−26 to 3]	−20 [−100 to 0]	0.24
Change at M6	−39 [−99 to −16] *	−32 [−100 to 0] *	0.94

Results expressed as mean and standard deviation or median [interquartile range] according to statistical distribution. W6: results at 6 weeks, M6: results at 6 months. * indicates *p* < 0.05 for within-group differences, i.e., between baseline and W6 or M6.

**Table 4 jcm-13-03318-t004:** Change (%) in range of motion and disability at 6 weeks and 6 months.

	Concentric Group	Eccentric Group	*p*-Value
Flexion OA knee (°)			
Baseline	136.6 ± 6.5	132.6 ± 7.4	
At W6	122.8 ± 43.8	99.0 ± 61.2	
Change at W6	0 [−4 to 0.7]	0 [−100 to 2] *	0.79
At M6	108.9 ± 58.0	98.2 ± 60.7	
Change at M6	0 [−4 to 1]	0 [−100 to 2] *	0.54
Extension OA knee (°)			
Baseline	−2.5 ± 3.5	−3.8 ± 4.3	
At W6	−2.3 ± 3.3	−2.3 ± 3.7	
Change at W6	0 [−71 to 0]	−42 [−100 to −13] *	0.34
At M6	−2.0 ± 2.8	−2.4 ± 3.8	
Change at M6	−21 [−85 to 0]	−75 [−100 to −33]	0.28
VAS OA knee (/100)			
Baseline	25.2 ± 17.1	35.5 ± 26.6	
At W6	17.3 ± 14.5	21.8 ± 22.1	
Change at W6	−17 [−75 to 0]	−41 [−100 to −17] *	0.31
At M6	18.0 ± 16.1	17.5 ± 20.1	
Change at M6	−40 [−100 to 0]	−76 [−100 to −33] *	0.44
WOMAC A pain (/20)			
Baseline	5.2 ± 2.6	6.8 ± 3.6	
At W6	4.2 ± 2.7	4.1 ± 4.0	
Change at W6	−10 [−50 to 0]	−29 [−100 to 0] *	0.30
At M6	3.9 ± 2.8	4.5 ± 4.3	
Change at M6	−12 [−50 to 0] *	−29 [−100 to 0] *	0.43
WOMAC B disability (/68)			
Baseline	13.7 ± 8.7	22.7 ± 13.1	
At W6	12.3 ± 8.8	14.1 ± 15.2	
Change at W6	−8 [−29 to 6]	−38 [−100 to 0] *	0.12
At M6	13.5 ± 11.0	14.8 ± 15.1	
Change at M6	0 [−22 to 25]	−20 [−100 to 0] *	0.11
WOMAC C stiffness (/8)			
Baseline	2.7 ± 1.5	3.8 ± 1.9	
At W6	2.3 ± 1.3	2.3 ± 2.4	
Change at W6	−20 [−33 to 0]	−25 [−100 to 0] *	0.35
At M6	2.1 ± 1.4	2.5 ± 2.1	
Change at M6	−50 [−75 to 0]	−20 [−100 to 0] *	0.87
10 m walk speed (m/s)			
Baseline	2.0 ± 0.2	1.9 ± 0.3	
At W6	1.8 ± 0.7	1.4 ± 0.9	
Change at W6	0 [−7 to 5]	−2 [−100 to 4] *	0.32
At M6	1.6 ± 0.8	1.4 ± 0.9	
Change at M6	0 [−16 to 3]	−6 [−100 to 2] *	0.54
200 m walk speed (s)			
Baseline	112.9 ± 17.3	119.7 ± 15.7	
At W6	98.4 ± 37.4	81.9 ± 51.0	
Change at W6	−3 [−8 to 0]	−5 [−100 to −1] *	0.56
At M6	87.3 ± 47.7	86.0 ± 53.4	
Change at M6	−6 [−13 to −1] *	0 [−100 to 7] *	0.15

Results expressed as mean and standard deviation or median [interquartile range] according to statistical distribution. W6: results at 6 weeks; M6: results at 6 months. * indicates *p* < 0.05 for within-group differences, i.e., between baseline and W6 or M6.

## Data Availability

The datasets analyzed during the current study and statistical code are available from the corresponding author on reasonable request, as is the full protocol.

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
