# Peer review of "Eccentric Muscle Strengthening Using Maximal Contractions Is Deleterious in Knee Osteoarthritis: A Randomized Clinical Trial"

_jcm, 2024, doi:10.3390/jcm13113318_

Round 1

Reviewer 1 Report

Comments and Suggestions for Authors

I read the manuscript with interest.  The data does suggest eccentric muscle training has greater risk for injury in patients with knee arthritis. 

In the 5 patients who withdrew were additional studies done to assess injury?  MRI or plain films?  Were CK levels assessed in patients to see the effects of muscle training?

Were other labs assessed such as CRP, etc assess impact of exercises?

How were the eccentric exercises done?  Many descriptions suggest the eccentric exercises may be done under a weight bearing load such as standing while rising from a squat, etc?  Was there an increased load across the arthritic knees in the eccentric vs concentric groups?   If so, would this contribute to increased pain in eccentric groups?

What arthritis medications were being taken by the study participants?  Was this controlled for in the study?

If arthritis medication was used to control joint pain, did it have an effect on the observed outcomes?

Comments on the Quality of English Language

adequate

Author Response

Reviewer 1

I read the manuscript with interest.

The data does suggest eccentric muscle training has greater risk for injury in patients with knee arthritis. 

In the 5 patients who withdrew were additional studies done to assess injury?  MRI or plain films?  Were CK levels assessed in patients to see the effects of muscle training?

We did not any additional exam to the 5 patients as they were not able to continue the training due to muscle lesion. it was a clinical confirmation without MRI or plain films. We did not assess the CK levels in patients as we know that whatever the type of muscle strengthening, the CK increase.

We add some elements in results part.

Were other labs assessed such as CRP, etc. assess impact of exercises?

We did not practice other exam like CRP to see the impact of exercise, as increase muscle strength was the primary endpoint.

Manuscript was revised accordingly particularly in the discussion part

How were the eccentric exercises done?  Many descriptions suggest the eccentric exercises may be done under a weight bearing load such as standing while rising from a squat, etc?  Was there an increased load across the arthritic knees in the eccentric vs concentric groups?   If so, would this contribute to increased pain in eccentric groups?

The interventions consisted of 12 sessions of quadriceps and hamstring muscle strengthening of the OA knee using the Cybex HumacNorm® isokinetic dynamometer, that means that he patients were seated with 110° hip flexion and 90° knee flexion. It is secured to the back of the chair by a lap and shoulder belt. The thigh homolateral to the tested knee is secured to the seat by a strap. The dynamometer's center of rotation is positioned opposite the lateral condyle. The device for attaching the leg to the dynamometer, including an anti-spin module, is positioned just below the tibial tuberosity.

It allows a dynamic voluntary muscular contraction to take place at a constant angular speed thanks to a self-adapting resistance It allows a dynamic voluntary muscular contraction to take place at a constant angular speed thanks to a self-adapting resistance

Each session began with a warm-up on a static bike for 5 minutes at low rate and low load.. The concentric group performed 3 sets of 10 repetitions of extension/flexion at adapted effort at a speed of 60°/s, for a total of 30 repetitions. The eccentric group performed 3 sets of 10 repetitions of extension/flexion at a speed of 30°/s for a total of 30 repetitions. Each repetition was followed by a 20s rest and 1mn between sets.

The intensity was increased progressively over the sessions. The 1st session was per-formed at 60% of the initial maximal moment of force, and the intensity was increased by 10% at each session, up to the 6th session if tolerated. In the event of fatigue or muscle pain during a session, the intensity was set at the level of the previous session. We therefore chose to quantitatively progress the exercises by asking participants to develop maximal torque with no pain at each repetition.

What arthritis medications were being taken by the study participants?  Was this controlled for in the study?

We found no significant difference between the 2 groups for pain and for antalgics consumption at baseline: 8/19 (42.1%) for concentric group vs. 11/19 (57.9%) for eccentric group.

If arthritis medication was used to control joint pain, did it have an effect on the observed outcomes?

Patients were free to take paracetamol or non-steroidal anti-inflammatory drugs on request, but this was not monitored during the study.

We add this point to the limitation part

Reviewer 2 Report

Comments and Suggestions for Authors

This is an interesting experiment to compare the effects of ECC and CONC training on quadriceps muscle strength in KOA patients. There were no major flaws in the experimental design. However, it is suggested that the authors could add figures to present the data to enhance the visualization of the results. In addition, it is recommended that the authors add a comparison of outcome metrics before and after treatment in the same group of KOA patients. Finally, a table is recommended for all adverse events to elucidate specific details.

Author Response

Reviewer 2

This is an interesting experiment to compare the effects of ECC and CONC training on quadriceps muscle strength in KOA patients. There were no major flaws in the experimental design.

However, it is suggested that the authors could add figures to present the data to enhance the visualization of the results.

We thank the reviewer for the comment.

Accordingly, a figure was added in revised manuscript to describe the result for the primary endpoint.

In addition, it is recommended that the authors add a comparison of outcome metrics before and after treatment in the same group of KOA patients.

We thank the reviewer for the comment. Tables 2 and 3 were revised according to the reviewer’s comment. Results before (baseline) and after treatment (at 6 weeks and at 6 months) were added.

Finally, a table is recommended for all adverse events to elucidate specific details.

We add a Table for all adverse events.

Patient

randomisation

Events

#5

concentric group

PREMATE STOP

#25

concentric group

PREMATE STOP

#36

concentric group

INJURY M6

#8

eccentric group

PREMATE STOP

#10

eccentric group

INJURY W6

#23

eccentric group

INJURY W6

#26

eccentric group

INJURY W6

#31

eccentric group

INJURY W6

#35

eccentric group

INJURY W6

Concentric group

Eccentric group

p-value

Concentric OA knee

Quadriceps

                Baseline

                At W6

                Change at W6 (primary endpoint)

                At M6

                Change at M6

1.29 ± 0.47

1.41 ± 0.77

28 [-17 to 55]

1.26 ± 0.99

23 [-99 to 74]

1.07 ± 0.44

0.94 ± 0.70

10 [-100 to 52]

1.03 ± 0.83

16 [-100 to 74

0.34

0.74

Harmstring

                Baseline

                At W6

                Change at W6

                At M6

                Change at M6

0.84 ± 0.32

0.83 ± 0.42

14 [-12 to 44]

0.62 ± 0.48

-13 [-99 to 36]

0.76 ± 0.29

0.59 ± 0.39

-1 [-100 to 24]

0.59 ± 0.48

-3 [-100 to 29]

0.21

0.96

Ratio Harmstring/Quadriceps   

                Change at W6

                Change at M6

-10 [-26 to 3]

-39 [-99 to -16] *

-13 [-100 to 0]

-47 [-100 to -11] *

0.42

0.96

Eccentric OA knee

Quadriceps

                Baseline

                At W6

                Change at W6

                At M6

                Change at M6

2.04 ± 0.66

1.91 ± 1.04

0 [-24 to 30]

1.44 ± 1.11

-12 [-99 to 6] *

1.81 ± 0.70

1.41 ± 1.02

0 [-100 to 22]

1.43 ± 1.14

-1 [-100 to 38]

0.65

0.59

Harmstring

                Baseline

                At W6

                Change at W6

                At M6

                Change at M6

1.38 ± 0.49

1.12 ± 0.66

-17 [-45 to 0]

0.74 ± 0.64

-40 [-99 to 16] *

1.29 ± 0.61

0.86 ± 0.68

-13 [-100 to 13] *

0.78 ± 0.68

-40 [-100 to 20] *

0.99

0.85

Ratio Harmstring/Quadriceps

                Change at W6

                Change at M6

-8 [-39 to 2]

-35 [-99 to 12] *

-20 [-100 to 0]

-32 [-100 to 0] *

0.39

0.83

Concentric-eccentric OA knee

Quadriceps

                Change at W6

                Change at M6

28 [-17 to 55]

23 [-99 to 74]

0 [-100 to 22]

-1 [-100 to 38]

0.04

0.22

Harmstring

                Change at W6

                Change at M6

14 [-12 to 44]

-13 [-99 to 36]

-13 [-100 to 13] *

-40 [-100 to 20] *

0.04

0.42

Ratio Harmstring/Quadriceps

                Change at W6

                Change at M6

-10 [-26 to 3]

-39 [-99 to -16] *

-20 [-100 to 0]

-32 [-100 to 0] *

0.24

0.94

Results expressed as mean and standard-deviation or median [interquartile range] according to statistical distribution.

W6: results at 6 weeks

M6: results at 6 months

* indicates p<0.05 for within-group differences, i.e. between baseline and W6 or M6.

Concentric group

Eccentric group

p-value

Flexion OA knee (°)

                Baseline

                At W6

                Change at W6

                At M6

                Change at M6

136.6 ± 6.5

122.8 ± 43.8

0 [-4 to 0.7]

108.9 ± 58.0

0 [-4 to 1]

132.6 ± 7.4

99.0 ± 61.2

0 [-100 to 2] *

98.2 ± 60.7

0 [-100 to 2] *

0.79

0.54

Extension OA knee (°)

                Baseline

                At W6

                Change at W6

                At M6

                Change at M6

-2.5 ± 3.5

-2.3 ± 3.3

0 [-71 to 0]

-2.0 ± 2.8

-21 [-85 to 0]

-3.8 ± 4.3

-2.3 ± 3.7

-42 [-100 to -13] *

-2.4 ± 3.8

-75 [-100 to -33]

0.34

0.28

VAS OA knee (/100)

                Baseline

                At W6

                Change at W6

                At M6

                Change at M6

25.2 ± 17.1

17.3 ± 14.5

-17 [-75 to 0]

18.0 ± 16.1

-40 [-100 to 0]

35.5 ± 26.6

21.8 ± 22.1

-41 [-100 to -17] *

17.5 ± 20.1

-76 [-100 to -33] *

0.31

0.44

WOMAC A pain (/20)

                Baseline

                At W6

                Change at W6

                At M6

                Change at M6

5.2 ± 2.6

4.2 ± 2.7

-10 [-50 to 0]

3.9 ±2.8

-12 [-50 to 0] *

6.8 ± 3.6

4.1 ± 4.0

-29 [-100 to 0] *

4.5 ± 4.3

-29 [-100 to 0] *

0.30

0.43

WOMAC B disability (/68)

                Baseline

                At W6

                Change at W6

                At M6

                Change at M6

13.7 ± 8.7

12.3 ± 8.8

-8 [-29 to 6]

13.5 ± 11.0

0 [-22 to 25]

22.7 ± 13.1

14.1 ± 15.2

-38 [-100 to 0] *

14.8 ± 15.1

-20 [-100 to 0] *

0.12

0.11

WOMAC C stiffness (/8)

                Baseline

                At W6

                Change at W6

                At M6

                Change at M6

2.7 ± 1.5

2.3 ± 1.3

-20 [-33 to 0]

2.1 ± 1.4

-50 [-75 to 0]

3.8 ± 1.9

2.3 ± 2.4

-25 [-100 to 0] *

2.5 ± 2.1

-20 [-100 to 0] *

0.35

0.87

10 m walk speed (m/s)

                Baseline

                At W6

                Change at W6

                At M6

                Change at M6

2.0 ± 0.2

1.8 ± 0.7

0 [-7 to 5]

1.6 ± 0.8

0 [-16 to 3]

1.9 ± 0.3

1.4 ± 0.9

-2 [-100 to 4] *

1.4 ± 0.9

-6 [-100 to 2] *

0.32

0.54

200m walk speed (s)

                Baseline

                At W6

                Change at W6

                At M6

                Change at M6

112.9 ± 17.3

98.4 ± 37.4

-3 [-8 to 0]

87.3 ± 47.7

-6 [-13 to -1] *

119.7 ± 15.7

81.9 ± 51.0

-5 [-100 to -1] *

86.0 ± 53.4

0 [-100 to 7] *

0.56

0.15

Results expressed as mean and standard-deviation or median [interquartile range] according to statistical distribution.

W6: results at 6 weeks

M6: results at 6 months

* indicates p<0.05 for within-group differences, i.e. between baseline and W6 or M6.

Round 2

Reviewer 2 Report

Comments and Suggestions for Authors

The authors have essentially addressed the issues I raised. I agree to publish the manuscript to the journal.